# Multi-Omics Endotypes in ICU Sepsis-Induced Immunosuppression

**DOI:** 10.3390/microorganisms11051119

**Published:** 2023-04-25

**Authors:** Alexis Garduno, Rachael Cusack, Marc Leone, Sharon Einav, Ignacio Martin-Loeches

**Affiliations:** 1Department of Clinical Medicine, Trinity College, University of Dublin, D02 PN40 Dublin, Ireland; 2Department of Intensive Care Medicine, St. James’s Hospital, James’s Street, D08 NHY1 Dublin, Ireland; 3Department of Anesthesia, Intensive Care and Trauma Center, Nord University Hospital, Aix Marseille University, APHM, 13015 Marseille, France; 4General Intensive Care Unit, Shaare Zedek Medical Center, Jerusalem 23456, Israel; 5Faculty of Medicine, Hebrew University, Jerusalem 23456, Israel

**Keywords:** sepsis, septic shock HLA-DR, immunosuppression, ICU, endotypes, gene modular features, transcriptomics, endothelial damage, regulatory inference, immunomodulation therapies

## Abstract

It is evident that the admission of some patients with sepsis and septic shock to hospitals is occurring late in their illness, which has contributed to the increase in poor outcomes and high fatalities worldwide across age groups. The current diagnostic and monitoring procedure relies on an inaccurate and often delayed identification by the clinician, who then decides the treatment upon interaction with the patient. Initiation of sepsis is accompanied by immune system paralysis following “cytokine storm”. The unique immunological response of each patient is important to define in terms of subtyping for therapy. The immune system becomes activated in sepsis to produce interleukins, and endothelial cells express higher levels of adhesion molecules. The proportions of circulating immune cells change, reducing regulatory cells and increasing memory cells and killer cells, having long-term effects on the phenotype of CD8 T cells, HLA-DR, and dysregulation of microRNA. The current narrative review seeks to highlight the potential application of multi-omics data integration and immunological profiling at the single-cell level to define endotypes in sepsis and septic shock. The review will consider the parallels and immunoregulatory axis between cancer and immunosuppression, sepsis-induced cardiomyopathy, and endothelial damage. Second, the added value of transcriptomic-driven endotypes will be assessed through inferring regulatory interactions in recent clinical trials and studies reporting gene modular features that inform continuous metrics measuring clinical response in ICU, which can support the use of immunomodulating agents.

## 1. Introduction

Despite decades of research for more effective cures, the incidence of sepsis and its mortality rate (26.7%) have remained essentially unchanged based on a pooled 21 study meta-analysis [1]. Transcriptomic data in sepsis have not been fully validated clinically as potential biomarkers for leading appropriate treatment applications. While there are many studies citing Phase 1 and Phase 2 clinical trials, murine and mouse models, and multi-center prospective cohort studies using machine learning to determine appropriate immunological endotype classifiers, it is important to note that these studies are still in the early stages of development and require further validation [2,3,4].

However, the potential of using transcriptomic data to improve sepsis diagnosis and treatment is an active area of research. By analyzing the gene expression patterns in blood samples, researchers hope to identify molecular signatures that can indicate the presence of sepsis or predict a patient’s response to treatment. This could ultimately lead to more targeted and effective treatments for sepsis patients. While there is still much work to be done before transcriptomic data can be widely used in clinical settings, the research in this area is promising and could have significant implications for sepsis diagnosis and treatment in the future [5,6,7,8,9].

Clinical criteria remain the gold standard for diagnosing sepsis, and there is no alternative that can replace them entirely. However, there is a growing interest in using omics- level data, such as transcriptomic data, to improve our understanding of the immunological dysregulation that underlies sepsis. One potential application of omics-data in sepsis is to develop new models that integrate this data with clinical criteria to improve diagnosis and treatment. For example, machine learning algorithms could be used to identify patterns in gene expression data that are associated with specific clinical outcomes or treatment responses. These models could then be used to guide personalized treatment decisions for sepsis patients, as reported in Sganzerla Martinez et al., evaluating subsets through interpretable Artificial Neural Networks in sepsis [10]. Notably, Bai et al., identified clinical phenotypes with differential responses of treatment in sepsis-associated acute respiratory distress syndrome (ARDS) using machine learning models [11].

Another potential application of omics-data is to identify new targets for sepsis treatments. By analyzing gene expression data, researchers can identify pathways that are dysregulated in sepsis and develop new therapies that target these pathways. This could lead to more effective and targeted treatments for sepsis patients. Overall, while there are limitations to using omics-data in sepsis, there is potential for these approaches to improve our understanding of the disease and inform targeted treatments and prognosis. However, it is important to continue to validate these approaches and integrate them with clinical criteria to ensure their clinical utility.

The current narrative review seeks to highlight the application of multi-omics data integration to define endotypes in patients with sepsis. The challenges for overcoming sepsis are (i) inconsistent immune eligibility and outcome criteria, (ii) a simplistic approach translated into guidelines for initial treatment, (iii) unacceptable prolonged time to diagnosis, and (iv) lack of patient trajectory assessment systems for a personalized transcriptomic driven approach to treatment, especially for a long-term perspective and addressing inhibition of certain inflammatory regulatory pathways. Classifying endotypes in critically ill patients with an infection in this review is defined as a link to a single molecular mechanism and those that share etiological and pathogenic pathways with nonlinear dynamic interactions.

## 2. Multi-Omics Single-Cell Data Integration for Immunological Profiling

Several recent investigations have led to the use of transcriptional data to determine differential gene expression profiles for both prognostic and pathway enrichment in complex disease leading to chronic inflammation. This has enabled the identification of immune cells with an active gene regulatory network at the single cell level. In the case of sepsis, quantifying a single ‘ome’ in a particular tissue would yield insufficiently granular details about its disease trajectory and host immune response. Tyler et al. highlights that in order to address these shortcomings, integrative multi-omics approaches have been developed, yet few studies have integrated clinical metrics that could inform on the mechanisms leading to immune dysregulation and distinguish a set of classifiers that would allow to distinguish subtypes in sepsis [12].

Recent advancement with regard to immunological profiling of critically ill patients with sepsis has driven the application of single-cell level technology. Single cell level analysis allows to evaluate the activity of the gene set in each cell. It is also used to assess the developmental trajectories of cells based on the expression profiles of RNA and protein simultaneously in single cells allows determination of dysregulated host response in infection, as shown in (Figure 1). A common challenge in the classification of patients into endotypes using transcriptomic data is the lack of reproducibility among different cohorts. Lack of reproducibility can be driven by the recruitment of distinctive populations, limited data on stage of severity, and assessment of the therapeutic management prior to ICU admission.

HLA-DR, or human leukocyte antigen-DR, is a molecule that is expressed on the surface of certain immune cells, including monocytes, macrophages, and dendritic cells. HLA-DR performs a critical role in the immune system’s ability to recognize and respond to foreign invaders, such as bacteria and viruses. In sepsis, HLA-DR expression is often decreased, which is thought to reflect a state of immune suppression that can contribute to the development of sepsis and its associated complications [13]. This immune suppression is referred to as immune paralysis, and it can make it more difficult for the body to fight off the infection and can increase the risk of secondary infections and other complications [13,14]. Similarly, in macrophage activation-like syndrome (MALS), which is a rare but potentially life-threatening condition characterized by a dysregulated immune response, HLA-DR expression may also be decreased. This dysregulation can result in the overproduction of pro-inflammatory cytokines and other immune molecules, leading to widespread tissue damage and organ dysfunction [14]. Overall, monitoring HLA-DR expression can provide insights into the immune system’s ability to respond to infections and other challenges and may have implications for the diagnosis, treatment, and management of a range of conditions, including sepsis, MALS, and immune paralysis.

Previous studies aimed to identify sepsis endotypes by analyzing the behavior of monocyte HLA-DR (mHLA-DR) expression during the first week after sepsis onset. The results showed that two-thirds of septic patients exhibited low or decreasing mHLA-DR expression, while in the remaining patients, mHLA-DR expression increased. Bodinier et al. discovered that measuring mHLA-DR expression on the first and third day after sepsis onset is sufficient for early risk stratification of sepsis patients [15]. This finding may help clinicians identify patients who are at higher risk of developing complications and who may require more intensive treatment. Overall, this study highlights the importance of monitoring mHLA-DR expression in septic patients and suggests that this biomarker may be a useful tool for identifying sepsis endotypes and for early risk stratification [15]. However, further research is needed to validate these findings and to determine the clinical implications of mHLA-DR monitoring in sepsis management.

Grondman et al. revealed a clear distinction in MALS as compared to the sepsis immune paralysis state. Results determined that the decrease in CD4 T cells was even more pronounced in sepsis patients that were classified as immune paralysis [4]. This is significant clinically as it shows that not only the innate immune system is compromised due to low monocytic HLA-DR expression, but that adaptive immunity is impaired in these patients, reflecting two important aspects in the pathogenesis of sepsis-induced immune paralysis. Results also showed that the identification of a sepsis-specific monocyte cluster also correlated to higher clinical scores, such as Acute Physiology and Chronic Health Evaluation (APACHE II), Sequential Organ Failure Assessment (SOFA), and Charlson Comorbidity Index (CCI) and led to increased mortality in sepsis patients [16,17,18]. Underlying an immune endotype that can infer in prognosis [4].

Similarly, Leijte and colleagues evaluated mHLA-DR expression kinetics in 241 patients with septic shock with different primary site of infection and pathogens using an unsupervised clustering method. They determined there was no significant association between site of primary infection or causative pathogen to mHLA-DR level expression [19]. However, delayed improvement in outcome correlated to the decline in mHLA-DR expression. Those patients also had a higher risk for adverse outcome suggesting changes in mHLA-DR overtime are of clinical importance in septic shock recovery and innate immune activation [19].

### 2.1. The Identification of Deleterious Neutrophil States and Altered Granulopoiesis in Sepsis

An UK study identified two endotypes sepsis response signatures (SRS1 and SRS2) based on analysis of the transcriptome response to sepsis of leukocytes in peripheral blood [20]. Analysis of gene expression pathways between the two groups showed functional differences related to T-cells, apoptosis, necrosis, cytotoxicity, and phagocytosis. SRS1 was associated with higher early mortality (22% vs. 10% for SRS2) and higher illness severity. SRS1 also had an LPS tolerance expression profile that was not present in SRS2 and downregulation of human leukocyte antigen (HLA) and T-cell activating genes. Sepsis-associated genetic loci were found most frequently associated with expression pathways involving genes that were differently expressed according to SRS profile [20]. Identifying patients with this immunological profile would be important as they might benefit from immune booster therapy.

The identification of deleterious neutrophil states and altered granulopoiesis in sepsis has also served in establishing a cellular immunological basis for transcriptionally defined endotypes in sepsis. A single cell multi-omics study Kwok et al. identified an immature neutrophil population specific to the immunosuppressive nature of sepsis via granulopoiesis dysfunction. The analysis, comprising 26 sepsis patients, showed that an *IL1R2*+ neutrophil state was expanded in a transcriptomic sepsis endotype [21]. This endotype ‘SRS1’, which was previously validated in other cohorts, was associated with an increased mortality and enrichment in response to IL-1 pathway in mature neutrophils, marking IL-1 out as a potential target for immunotherapy in these SRS1 sepsis patients [21]. Even though past trials targeting IL-1 inhibition have failed to reduce mortality rates, it should be noted that these trials had the absence of patient stratification, which could have led to the low success rate. Let us take the re-analysis of a phase III trial of IL-1R blockade with anakinra; Shakoory et al. identified improved survival in subgroup of anakinra-treated patients with hepatobiliary dysfunction and disseminated intravascular coagulation [22]. This suggests a need to identify target patients prior to therapy implementation, as this study showed a promising outcome of IL-1 inhibition in SRS1 sepsis patients [22]. Trials administering GM-CSF are still ongoing and propose a potential immunotherapeutic target. This study revealed increased levels of G-CSF in SRS1 patients and enhanced functional capacity in regulating circulating immature and immunosuppressive cells [21,23].

### 2.2. Transcriptomic Driven Endotype in Sepsis Immunocompromised Patients

Sepsis and bloodstream infections remain a leading cause of death in immunocompromised patients with cancer and accounts for 5–6 million deaths of ~30 million cases per year worldwide [24]. Although sepsis-related mortality rates in cancer patients are decreasing over time, the incidence of cancer patients admitted to ICU almost half of admissions due to septic shock have dramatically increased, with long-term prognosis showing a higher 180-day mortality rate due to septic shock higher in cancer patients 51–68% compared to 44% in non-cancer patients [25]. Overall, patients with hematological malignancies, mainly acute leukemia pose a higher risk of developing sepsis [26]. Past research in myeloid-derived suppressor cells (MDSCs) both for cancer and sepsis has shown MDSCs expand during chronic and acute inflammatory conditions. In cancer, increased MDSCs induce detrimental immunosuppression, but little is known about the role of MDSCs after sepsis [27].

The presence of a systemic inflammatory response in patients with oncological disease (hematological/solid malignancies) warrants a risk stratification endotype that can characterize immune-related genes and immune microenvironment features in sepsis immunocompromised patients. A sc-RNA-seq study observing the differential effect of sepsis on innate versus adaptive immunity revealed T cell and monocyte-specific RNA distinctive transcriptomic profiles in septic and nonseptic critically ill patients and in patients with cancer [28]. Sepsis revealed a marked phenotypic shift toward downregulation of multiple immune response pathways in monocytes suggesting impaired immunity as compared to those patients with cancer. There was a pronounced effect at the gene transcription level in CD4 and CD8 T cells. Notably, the analysis identified potential mediators of sepsis-induced immunosuppression, including *Arg-1*, *SOCS-1,* and *SOCS-3*, which were highly upregulated in multiple cell types and negative costimulatory molecules, including *PD-1* and *CTLA-4* upregulated in sepsis [28].

In preclinical studies, administration of either *PD-1* or *PD-L1* antibody improved overall survival in the CLP mouse model of sepsis [29]. It has also been demonstrated that utilization of either *PD-1* or *PD-L1* antibodies in blood donor restores neutrophil and monocyte function and reverses T-cell exhaustion supporting that immunoadjuvant therapies that are effective in cancer may be a potential target in patients with sepsis. In animal models, sepsis-induced immune dysfunction has been shown to impact malignant growth, causing a post-septic immunosuppressive environment that may induce cancer recurrence in sepsis patients. In a study conducted by Vigneron et al., the experiment comprised of mice previously inoculated with MCA205 fibrosarcoma cells which were later subjected to septic challenges. There was a tumor growth inhibition followed by LPS challenge suggesting a regulatory role of Toll-like receptor (Tlr4) in this setting [30]. There was a low expression of MHC class 1 onto MCA205 cells suggesting the involvement of Natural Killer (NK) cells in sepsis-induced tumor inhibition. 

Septic insults applied to mice with cancer promoted the main anti-tumoral NK functions of IFNγ production and degranulation [29,30]. This further warrants a dual effect endotype model that considers anti-tumoral NK function and downregulation in cancer-related sepsis and the potential for allogenic NK cellular transfer therapy for sepsis insult. Endotypes in immunocompromised sepsis patients with cancer require a better understanding of the innate and adaptive immune cell functions to address the antitumoral mechanisms elicited by antibacterial responses. Only then, can we target antitumoral properties in these patients and develop potential immunomodulatory agents.

### 2.3. Transcriptomic Landscape of Chronic Critical Illness in Late Sepsis

Chronic critical illness in late sepsis is an ongoing adverse long-term risk. The host response trajectory in sepsis is classified as developing [31] an early multi-organ failure (MOF) response leading to death, ref. [32] patients with rapid recovery and [33] a maladaptive host response leading to chronic inflammation, T cell exhaustion, expansion of suppressor cell functions, and protein catabolism [34]. This third trajectory has been shown to further elicit a unique circulating leukocyte transcriptome in surviving septic patients. Darden and colleagues conducted a prospective cohort study of surgical ICU patients that incorporated a genome-wide expression analysis on total leukocytes in whole blood on day 1 of systemic inflammatory response and day 14 from sepsis survivors who rapidly recovered and revealed that although similar gene expression patterns were shown in CCI patients and those who survived there was still an exhibited differential expression of 185 unique genes [35] compared with rapid recovery patients.

Many genes in the study that were correlated with poor disposition have been indicated in other studies including *BLK*, *BAG6*, *FOXO4*, and *ERF* [20,35]. However, upregulation of these genes was associated with inflammatory response at day 14 with dismal outcome in patients. Other genes highlighted to be altered at day 14 post sepsis and CCI included *ATG12*, *EHD1*, *NACC1*, and *SLC7A5*, which were involved in host protective immunity, promotes autophagy, T cell differentiation, and stem cell self-renewal and maintenance [35]. These findings could inform which CCI post-sepsis patients may benefit from targeted immunotherapies and to further understand the molecular process behind some CCI patients rapidly recovering as compared to other they may require immunomodulation and specific timing of treatment, as shown in (Table 1), of past and active immunomodulation trials.

A single cell RNA-seq study analyzing non-myeloid circulating cells revealed a unique transcriptomic pattern of multiple immune cell subtypes, including B- and CD4+, CD8+, activated CD4+, activated CD8+ T- lymphocytes, natural killer (NK), NKT, and plasmacytoid dendritic cells in late sepsis patients [36]. Notably, these circulating lymphoid cells maintained a transcriptome that reflected persistent immunosuppression and low-grade inflammation that differed between patients with bacterial vs. fungal sepsis (i.e., higher expression of cytotoxic genes among CD8+ T lymphocytes in late bacterial sepsis [36]). This suggests that identifying this transcriptomic pattern in late sepsis in non-myeloid cell would allow to provide the appropriate immunomodulatory therapy to those patients revealing a host endotype response driven by persistent inflammation. Darden et al. was also determined to highlight the role of myeloid derived suppressor cells in late sepsis and transcriptional relevance clinically. Three subsets of MDSCs clusters, granulocytic (G-), monocytic (M-), and early (E-), were identified, and sepsis was associated with an increased relative expansion of G-MDSCs at 21 days [37]. However, the analysis demonstrated that CCI is partially dependent on the original septic insult and that the pilot data indicates a distinct immunosuppressive pathway in late sepsis that can be further targeted, especially in patients with cancer-related sepsis considering past studies investigating cancer-induced MDSC subsets and proliferation role.

## 3. Omics of Endothelial Dysfunction and Microcirculatory Alterations

Inflammatory signals result in leukocyte dysregulation and endothelial changes, such as increased permeability and cell migration [38]. Proteomics specifically provides analysis of transcriptional changes and interactions that connect genotype to phenotype (Table 2) [39]. Ricaño-Ponse et al. identified endotypes based on inflammatory proteins in moderately ill ICU patients with severe infections suggesting there should be consideration for inflammatory proteome profiling that can be used to stratify patients. Proteome profiling would allow the identification of patients with higher concentrations of inflammatory protein, which has been already correlated with hyperinflammation in patients who are older and clinically have lower lymphocyte counts in early sepsis [40]. Functional immune endotyping based on enzyme linked targets has also shown promise in the classification of early sepsis immune dysregulation trajectories. Measuring the number and intensity of cytokine-secreting cells, Mazer and colleagues revealed non-survivor septic patients had early, profound and sustained suppression of their innate and adaptive immune system. They also showed increased cytokine production compared with healthy controls consistent with either an appropriate or excessive immune response. IL-7 restored ex vivo IFN-g production in septic patients [41]. These non-survivor septic patients with excess suppression may benefit from adjustive immunotherapy targeting IFN-g production and restore innate response to infected secreting cells.

Endothelial cell dysfunction in sepsis combines intercellular junction breakdown, increased adhesion molecule surface expression, and stimulation of complement and coagulation cascades [42,43]. Endothelial omics provide a rational basis for understanding sepsis pathology and computer modelling relevant therapies based on phenotypes [44]. Using RNA sequencing to characterize the entire transcriptome of a single cell or mass spectrometry to describe the heterogeneity of cell responses by measuring proteins in samples, omics analyses large data sets to subtype groups and tailor treatments [45,46]. By identifying patient-specific biology in sepsis, computer modelling can then be used to translate this into a credible endotype and classify patients according to shared characteristics [45,47].

Both microbial products and cells involved in the inflammatory cascade can stimulate endothelial cells, causing a change in the cell to increase permeability and secretion. IL-6 is produced by the endothelial cell and leukocytes and decreases expression of intercellular junctional protein, and impairing barrier function [48]. IL-6 can act as both a pro- or anti-inflammatory cytokine depending on its source and timing relating to the immune response [49]. ICAM expression on the endothelial cell surface is important to initiate transmigration of leukocytes, shedding of adhesion molecules, such as ICAM, E-selectin, and VCAM, is associated with organ failure severity [50,51]. Sepsis deactivates the endothelial anticoagulant and fibrinolytic functions. Plasminogen activator inhibitor (PAI-1) is released by endothelial cells and monocytes. Expression of the gene coding PAI-1 is increased in inflammation. Blocking the effects have different effects depending on the source of sepsis. In LPS challenge, PAI-1 inhibition reduced mortality by reducing hypercoagulation, whereas, in a pneumonia model, PAI-1 deficiency attenuated neutrophil responsiveness in the lungs and increased mortality [52,53]. The endothelial response to infection and inflammation varies between organs and vascular beds. The lung endothelium transcriptome is dramatically affected by inflammation, possibly differentiating patients with respiratory sepsis from other sepsis patients [54]. This difference is due to local communication of the parenchyma with the endothelium, which changes how the endothelium across different organs reacts. Endothelial gene expression is reduced at early and late stages of inflammation in lung and brain tissue. Endothelial cells not only control the interactions of circulating molecules with the interstitium, but they also control the composition of the plasma. As inflammation continues, platelets and components of the coagulation cascade alter microcirculatory flow dynamics causing stagnation and activated cells changing the blood rheology [38,55]. The shear stress on the endothelium is important in maintaining the barrier integrity and leads to expression of different adhesion and signaling molecules encouraging protein binding [56].

### 3.1. Sepsis-Induced Metabolic Changes and Microcirculatory Damage

Sepsis induces metabolic changes in endothelial cells. Endothelial cells differ across organ beds and have different metabolic stress responses. Understanding these differences could lead to new ways to identify sepsis sub-phenotypes. Enhanced glycolysis in endothelial cells in sepsis increases lactate, which causes vascular leak by eroding the vascular-endothelial-cadherin (VE-cadherin) bonds by stimulating extracellular signal-regulated kinase-2 (ERK-2), resulting in a vicious cycle [57,58]. Sepsis also upregulates endothelial specific glycolysis genes, which, when inhibited in LPS-treated mice, reduced neutrophil infiltration, lactate production, and endothelial permeability [59,60]. Endothelial cells express four isotypes of nicotinamide adenine dinucleotide phosphate (NADPH) oxidases, which produce reactive oxygen species. The expression of each isoform changes over time following a septic insult resulting in loss of tight junction proteins and barrier dysfunction due to oxidative stress [57]. These oxidative genes are responsible for endothelial cell apoptosis, trans-endothelial migration, and Toll-like receptor 4 transcription [61,62]. Reactive oxygen species activate sheddase enzymes that expose ectodomains of transmembrane proteins leading to coagulation activation, leukocyte adhesion, and vascular permeability [63]. Fatty-acid oxidation and transport can also increase endothelial permeability and apoptosis in sepsis, though further research in this area is warranted [64]. Arginine is an amino acid that can be transformed into nitric oxide, which is involved in immunity as well the vasodilatory response that results in hypotension in sepsis. Arginine can balance the Ang/Tie-2 pathway to support endothelial barrier function, and arginase-1 is also protective for endothelial cells [65]. Arginase deficient mice experience much worse microcirculation damage following LPS injection, meaning it could represent an important future target [66].

Increased concentrations of endothelial components in plasma correlate with severity of microvascular dysfunction and sepsis. Non-survivors consistently have higher concentrations of circulating glycosaminoglycans in sepsis [64]. The enzymes responsible for cleaving endothelial components have been proposed as biomarkers for sepsis as they are correlated with outcome and severity [67]. Hyaluronic acid and syndecan are both raised in non-survivors in sepsis [68,69]. Endocan is a proteoglycan that has been shown to rise with ICU admission, to be related to length of stay, response to therapy and outcome [70], [71].

Alteration of the normal microcirculation has been identified as a defining feature of the translation from sepsis to septic shock [72,73]. Tissue oxygenation and cell metabolism is affected by cytokine mediated inflammation, coagulation cascade activation and glycocalyx shedding that affect microcirculatory flow. Many studies have aimed to target the microcirculation as a prognostic indicator and therapeutic target for patients with sepsis [74,75]. Handheld vital microscopy has been used to assess the microcirculation function at the bedside, but there is a lack of biomarker correlation to microcirculation dysfunction [76,77]. Breakdown of the endothelial glycocalyx is an important pathophysiological step in sepsis and septic shock. The endothelium lines the vasculature and is covered by a jelly-like layer of proteoglycans and glycosaminoglycans that are intimately involved in cell signaling, propagation of inflammation, and coagulation [42,78]. In critical illness, inflammatory signals cause weakening of adherens and tight junctions in the endothelium causing paracellular gaps to allow molecules and fluid to escape into the interstitium as demonstrated in (Figure 2) [79,80].

### 3.2. Biomarkers of Endothelial Damage to Assess Treatment Response in Critical Illness

Biomarkers of endothelial damage and endothelial activity can be used to identify at risk patients and prognosticate in critical illness [81,82]. The challenge is to identify endothelial damage with optical assessment and translation of available biomarkers. Studies trying to identify microcirculation targets in patients with sepsis have found that the microcirculation changes over time, meaning that the first 48 h of resuscitation treatments have different effects [74]. Microcirculation differences have been proposed as a way to stratify patients in ICU [83]. Heterogeneity of results in trials aiming to find treatments for microcirculation dysfunction indicates the possibility of different mechanisms underlying the damage [84].

Different pathways leading to endothelial damage could affect the microcirculation at different timepoints or in different organs, leading to heterogeneity and the possibility of group stratification. Different phenotypes of sepsis patients could have microcirculation differences that respond differently to treatments. Assessing the change in microcirculation during sepsis could help enrich studies, prognosticate for patients, and identify which patients will benefit from treatments [85,86]. Endothelial disruption causes organ dysfunction because of changes in cellular metabolism and homeostasis. Lactate is a product of maladapted metabolism that has been used to identify sepsis patients for decades.

Adrenomedullin is an amino acid preprohormone that is cleaved into two biologically active compounds, adrenomedullin and pro-adrenomedullin (pro-ADM) [87]. Although this molecule is widely available, it has specific receptors on the endothelium and mediates a range of functions, such as vascular tone, fluid, and electrolyte homeostasis [88,89]. Adrenomedullin is vital in the creation and maintenance of endothelial barrier function, demonstrated in multiple murine knock-out models of either adrenomedullin or its receptor [90]. Inflammatory cytokines, such as tissue necrosis factor (TNF) and interleukins (IL) -1α, 1β, or lipopolysaccharide (LPS) found on Gram-negative bacteria have been shown to increase expression of adrenomedullin from fibroblasts, macrophages, and vascular smooth muscle cells [91,92]. Sepsis from various sources causes adrenomedullin release from the cells in either the small intestine or the lung [93,94]. Adrenomedullin acts by stabilizing inter-endothelial cell junctions, reducing permeability [95].

Due to its short half-life, adrenomedullin itself had little use as a biomarker; however, midregional proADM (MRproADM) is present in large quantities in the blood of sepsis patients and is stable enough to be measured clinically [96]. Elevated MRproADM concentrations correlate with clinical scores of severity in ICU patients, such as sepsis-related organ failure assessment (SOFA), and acute physiology and chronic health evaluation score (APACHE), and inflammatory biomarkers, such as C reactive protein (CRP), IL-6, IL-10, procalcitonin (PCT), and TNFα [97]. Studies have shown MRproADM to provide accurate stratification of ICU patients in terms of severity and prognosis [98,99]. The trend of MRproADM over the course of ICU in response to treatment accurately identifies patients at risk of worse outcomes better than other markers [99]. Multivariate analysis of sepsis patients found that MRproADM can predict five different causes of organ failure respiratory, coagulation, cardiovascular, neurological, and renal better than other biomarkers, meaning it could become important in diagnosis and prognosis [100].

Therapeutically, non-neutralizing ADM-binding Ab adrecizumab has also been shown to inhibit the production of pro-inflammatory cytokines, thereby reducing organ dysfunction, vascular leakage and the need for vasopressor treatment in various murine sepsis models [101,102]. Evidence suggests that adrecizumab increases ADM plasma levels by shifting ADM from the interstitium to the blood, thereby reducing vascular smooth muscle cell vasodilation and endothelial dysfunction [103]. There is a need for its application in clinical trials to assess the use of adrecizumab to increase the ratio of ADM in the circulation vs. interstitium. Phase II clinical trial of patients with septic shock with elevated concentrations of ADM are underway [104].

MRproADM is not the only marker of endothelial damage that can be used in ICU patients. Endothelial damage is an important pathophysiological mechanism that can be evaluated to separate patients into homogenous groups in ICU. Syndecan-1 is a component of the endothelial glycocalyx and family of syndecan molecules [105,106]. It is one of the glycocalyx signaling molecules that is constantly generated and shed; soluble plasma concentrations have been investigated in the context of sepsis and septic shock as shedding is initiated by leukocyte derived proteases [107]. A systematic review of glycocalyx degradation and sepsis outcomes in ICU patients found elevated blood concentrations of syndecan-1 in patients who developed multiorgan dysfunction, renal failure, or non-survivors [108]. Syndecans are found in wound fluid and peripheral blood of patients with critical illnesses and are involved in vascular permeability, endothelial damage, and disassembly of cell junctions in response to hypoxia and TNFα [105,109,110].

Angiopoietin-2 (Ang-2) action on the endothelium is in opposition to Ang-1, mediated by the soluble thrombomodulin and soluble angiopoietin receptor (TIE-2) receptor [111,112]. Ang-2 induces permeability and is raised as a marker of glycocalyx damage, correlating with sublingual measures of microvessel injury clinically [113]. The MYSTIC study showed glycocalyx damage and microcirculation derangement is associated with raised Ang-2 concentrations in sepsis [114]. Ang-2 also increases with increasing severity of sepsis, showing that septic shock patients can be discriminated using this biomarker of microvascular damage [115]. Vascular endothelial growth factor (VEGF) is expressed on the endothelium and is associated with angiogenesis and microvascular permeability, causing oedema and hypotension in sepsis [116]. Ang-2 and Vascular endothelial growth factor (VEGF) were two biomarkers assessed as part of the Protocolized Care for Early Septic Shock (ProCESS) randomized controlled multicenter trial [117]. At baseline and 24 h, Ang-2 and VEGF were associated with increased 60-day mortality in 1341 patients [118]. To date, there is no definitive clinical evidence that supports the use of Ang/Tie2 modulators in sepsis for routine clinical use.

## 4. Gene Modular Features Informing Clinically Relevant Endotypes

### 4.1. Hepatic Injury Dysregulation in Sepsis

Considering sepsis has a multi-factorial origin, there is a need to identify endotypes driven by gene modular features. During hepatic injury, monocyte-to-macrophage differentiation is a key event because it results in the dysregulation of immune response; limiting the host’s ability to control appropriate innate response. This is a relevant issue as over 40% of liver injuries in sepsis are due to pathogen resistance and tissue damage [119,120]. Asialoglycoprotein receptor 1 (ASGR1) is known to be enriched in classical monocytes of peripheral blood mononuclear cells (PBMCs). In recent studies, *ASGR1* target receptor gene was shown to promote monocyte-to-macrophage differentiation via up-regulating CD68, F4/80, and CD86. It also acts as a suppressive factor, downregulating the level of IL-1β, IL-6, and TNF-α; alleviating liver injury; and improving survival after sepsis [120]. Modulating gene features of *ASGR1* can serve as a potential for liver injury reconstruction by classifying patients who develop resistant and tolerogenic response patterns. Its target would allow it to balance the host defense (i.e., pathogen killing, clearing, and organ injury). Deciphering this molecular mechanism across patient cohorts and its correlation to MOF and failure of cellular communication in the liver would improve poor prognosis and liver dysfunction by restoring tolerogenic signaling and regeneration [120]. Notably, another target that can improve liver injury in sepsis patients is one that can inhibit immunomodulatory pathways. A blockade of Sodium Taurocholate Cotransporting Peptide (NTCP) to reduce bile acid absorption, proapoptotic stimuli, and oxidative stress in hepatocyte [121].

### 4.2. Sepsis Induced Cardiomyopathy (SICM) Gene Modular Features

Single cell ‘omic’ technologies that develop gene modulating models also have relevance in cases where patients develop sepsis induced cardiomyopathy (SICM). Currently, there are minimal therapeutic strategies that are targeted at patients with SICM with poor prognosis. However, there is research that has assessed beta blocker efficacy in sepsis and its immunological effects. Administration of landiolol was shown to prevent cardiac dysfunction in ovariectomized females with an overexpression of JAK2, natriuretic peptide receptor 3, phosphorylated-AKT:AKT, and endothelial nitric oxide synthase cascade [122]. It was also highlighted that indexed end-diastolic volume reduction after sepsis was correlated to the activation of calcium influx pathways whole in control patients there was down-regulation of these pathways [122].

Notably, Zhou et al. investigated the protective role of the hydrogen sulfide (H2S) donor GYY4137 via the NLRP3 pathway [123]. RNA expression profiles revealed a crosstalk between macrophages and cardiomyocytes with lower serum H2S levels and heart dysfunction. It was noted that GYY4137 reduced macrophage infiltration in septic heart tissue and that co-stimulatory genes were involved in the inflammatory process of these patients. GYY4137 effectively inhibited the NLRP3 inflammasome pathway and its activity in regulating macrophage populations. There was also a clear reduction in secretion of inflammatory factors and decreased the production of ROS (imbalance) in cardiomyocytes [123]. Suggesting GYY4137 has a protective effect against developing a circulatory disturbance in SICM patients. Clinically this poses a potential target as the findings showed that H2S could protect the heart by lowering oxidative stress. The development of SICM is primarily characterized by macrophages infiltrating the heart. An NRLP3-mediated inflammatory response occurs in these macrophages, releasing multiple pro-inflammatory factors and causing ROS accumulation in cardiomyocytes. This results in cardiac dysfunction and sepsis immunosuppression in patients with active infiltrating macrophages [123].

The pathogenesis of SICM is very complex, which is why recent efforts have tried to integrate bioinformatic pipelines analyzing GEO data to identify potential biomarkers for septic cardiomyopathy subclassifications. Pu et al. targeted a severe threat in septic cardiomyopathy progression by conduction and enrichment analysis with the intersection of differentially expressed genes and co-expression network analysis [124]. Results identified 479 potential target genes in association with SICM severity. The enrichment pathway analysis was shown to be involved in the positive regulation of protein kinase A signaling and T cell receptor signaling [124]. Notably, *APEX1* was significantly upregulated in the SICM group with higher mortality rate and showed a significantly adjusted FDR value in signaling pathways, driving base excision repair and glycosaminoglycan biosynthesis. Therefore, *APEX1* may be a valuable biomarker to determine the pathobiological and immune response signals in patients at risk of SICM. This pathway could also help find suitable treatment for these patients.

## 5. Continuous Metrics Measuring Endotypes and Clinical Response in ICU

### 5.1. Transcriptomic and Immunological Metrics Guiding Immunotherapy

Continuous transcriptomic and immunological metrics are used to assess immune activation and guide immunotherapy in sepsis and septic shock. These metrics can also serve to determine risk profiles of critically ill patients for CCI after sepsis. After diagnosis, treatments are administered to mitigate the progression of sepsis towards septic shock and multiple organ failure (MOF). The lack of recognition has been lost in translation when interpreting new accepted guidelines for sepsis. Starting antibiotic therapy during the first hours has been shown to significantly improve prognosis. Despite adequate antibiotic administration, bacterial infection scores and mortality rates did not decline over time [125].

Scicluna et al. described four different endotypes upon ICU admission for septic patients using a discovery and validation cohort under the acronym of Molecular Diagnosis and Risk Stratification of Sepsis (MARS) Consortium [126]; specifically, into endotypes are most helpful providing a basis for therapeutic rationale. Low mortality groups in these studies appear to be described by a more active adaptive immune response, whereas high mortality endotypes have more depressed immune function [20,126,127,128,129]. This holds for true SRS 2 vs. SRS 1, MARS 3 vs. MARS 4, adaptive vs. inflammopathic, and α vs. δ and subclass A.

In the Sweeney et al. study, previously validated inflammopathic, adaptive, and coagulopathic sepsis endotypes were incorporated in a COVID-19 (viral sepsis) cohort to determine the parallel of these predefined 33-messenger RNA endotypes classifier recapitulated immune phenotypes and how these could inform current interaction to patients receiving immunomodulatory therapy [2]. Notably, all death occurred in patients with near-zero probability of the adaptive endotype [2]. The assigned classes were comparable to prior study showing 44% classified as an adaptive endotype suggesting that the host response of clinical ‘sepsis,’ whether it bacterial or viral with secondary infection, may have similar pathways across infection types, despite known different host response trajectories.

Notably, Nusshag et al. also addressed in a cohort study how transcriptomic endotypes could classify a unique risk profile in those sepsis patients with acute kidney injury (AKI) [130]. The secondary analysis comprised of 200 adult patients. Results indicated that the patients with the inflammopathic classification exhibited the highest need for renal replacement therapy (RRT). These patients had an early elevation in mean lactate levels plasma concentration and delayed mortality. In contrast, patients classified as developing the coagulopathic dysregulation, despite exhibiting high incidence of persistent AKI, showed the lowest incidence of all clinical outcomes suggesting the need for incorporating transcriptomic assessment early in sepsis to stratify renal risk [130].

The PROVIDE randomized clinical trial identified four levels of immunoparalysis of sepsis using ferritin and HLA-DR. Macrophage-activation-like-syndrome (MALS) is associated with plasma ferritin concentration above 4420 ng/mL, whereas immunoparalysis is characterized by less than 5000 HLA-DR receptors on CD14 monocytes [131]. Anakinra treatment targeting MALS was shown to improve 7-day outcomes with a decrease in SOFA score. More precisely, patients with MALS were noted to have a lower absolute count of platelets and prolonged activated partial thromboplastin time (aPTT), and higher levels of aspartate aminotransferase (AST), alanine aminotransferase (ALT), and total bilirubin [131]. These characteristics signified a positive treatment effect in the early stage of sepsis insult but did not improve day 28, and treatment was stopped early due to a hyperinflammatory effect. However, as the PROVIDE classification of immunoparalysis immunotype parallels with the MARS 1 endotypes, further trials are warranted to optimize treatment that targets T cell function.

### 5.2. Pharmacologic Immune Modulation in Sepsis

Steroids are a class of drugs that have anti-inflammatory and immunosuppressive effects. They are widely used in clinical practice to treat a variety of conditions, including autoimmune diseases, allergic reactions, and certain types of cancer. Steroids work by binding to specific receptors in the body, such as the glucocorticoid receptor (GR), and modulating gene expression and cellular signaling pathways. This can lead to a decrease in the production of pro-inflammatory cytokines and other immune molecules and a decrease in the activity of immune cells, such as T cells and B cells [132]. In the context of community-acquired pneumonia (CAP), steroids have been studied as a potential adjunctive treatment to antibiotics, with the aim of reducing inflammation and improving outcomes [132,133]. Several clinical trials have investigated the use of steroids in CAP, and while the results have been mixed, some studies have suggested that steroids may have a beneficial effect in certain patient populations [134]. A recent review paper analyzed systematically all published randomized controlled trials involving patients with severe CAP found that the use of steroids was associated with a significant reduction in mortality and length of hospital stay and a decreased risk of treatment failure [134]. However, it should be noted that the benefits of steroids were primarily observed in patients with severe CAP and those who required mechanical ventilation or vasopressor support. Overall, while steroids may have a role in the management of severe CAP, their use should be considered on a case-by-case basis, considering the potential risks and benefits. It is important to note that steroids can have significant side effects, such as increased risk of infection, hyperglycemia, and gastrointestinal bleeding, among others [133,134].

Moreover, a good amount of research has been conducted to determine the adjunctive effect of steroids in patients with sepsis. Glucocorticoids appear to have beneficial effects in some patients and cause harm in others [135,136]. The timing and dose of glucocorticoids also performs a role, as low dose infusion of hydrocortisone improves hemodynamic profile in sepsis, possibly as a result of nitric oxide inhibition [137]. However, the immunomodulatory effects of glucocorticoids should also be considered. Hydrocortisone treatment affects both inflammatory and anti-inflammatory responses, so may be more effective early in shock when there is a predominantly inflammatory response, avoiding the later stages of sepsis when there is more immunosuppression. Re-examining the patients in the Vasopressin vs. Norepinephrine as Initial Therapy in Septic Shock (VANISH) trial found that hydrocortisone was harmful to those patients in the immunocompetent SRS 2 endotype [135]. Response to or benefit from glucocorticoids can be assessed by measuring cytokine profiles [138]. There is also evidence of differential responses of patient immune systems to sepsis, with delineation of a subset of patients with lower endogenous immunoglobulin levels associated with mortality [139]. Studies looking for biomarker signatures of sepsis and septic shock have shown multiple combinations of immune markers change in sepsis and septic shock.

The use of immunoglobulin therapy for sepsis is still a matter of debate, and there is no clear consensus on the type, dosing and timing. The most used types of immunoglobulins for sepsis are intravenous immunoglobulin (IVIG) and IgM-enriched formulations. The optimal dose of IVIG for sepsis is still unclear. Some studies have used a dose of 0.5–1 g/kg, while others have used higher doses (up to 2 g/kg) [140]. The dose may also depend on the patient’s age, comorbidities, and the severity of sepsis. The optimal timing of IVIG administration for sepsis is also unclear. Some studies have suggested that early administration of IVIG (within the first 24–48 h of sepsis) may be more effective, while others have shown benefit with later administration (after 48 h) [140].

The use of IVIG for sepsis is still controversial, and more research is needed to determine its safety and efficacy. Additionally, IVIG therapy is associated with an increased cost and may not be readily available in all countries. The stratification of patients depending on their immunological response to sepsis could be an important way to determine who should receive glucocorticoid or immunoglobulin therapy.

Other novel therapies that have shown some promising potential benefits are immunoglobulins, non-neutralizing ADM-binding Ab adrecizumab [101,102] has also been shown to inhibit the production of pro-inflammatory cytokines, monoclonal antibodies targeting checkpoint molecules, such as programmed-death 1 protein (PD-1) and its ligand *PD-L1*, molecules involved in the modulation of *HLA-DR* level as a potential target regulating anti-infectives [141,142].

## 6. Conclusions

The diagnostics and therapeutic interventions adopted to detect and manage sepsis are driven by changes in symptoms related to organ dysfunction so that timely diagnostic and interventions remain a challenge. Generating predictions on patient outcomes based on multiple transcriptomic data outputs can provide a better characterization of the patients. Modeling the regulatory interactions that include biomarkers with known medical, biological, and immunological activity are a need in treating critically ill patients with sepsis more effectively.

## Figures and Tables

**Figure 1 microorganisms-11-01119-f001:**
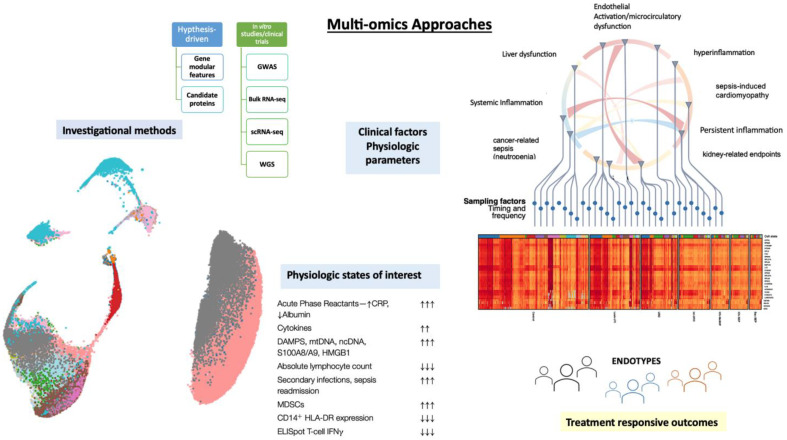
Conceptualizing a new model for understanding mechanisms and modulation involved in sepsis-induced dysfunctions detailing the physiologic states of interest from the recent literature.

**Figure 2 microorganisms-11-01119-f002:**
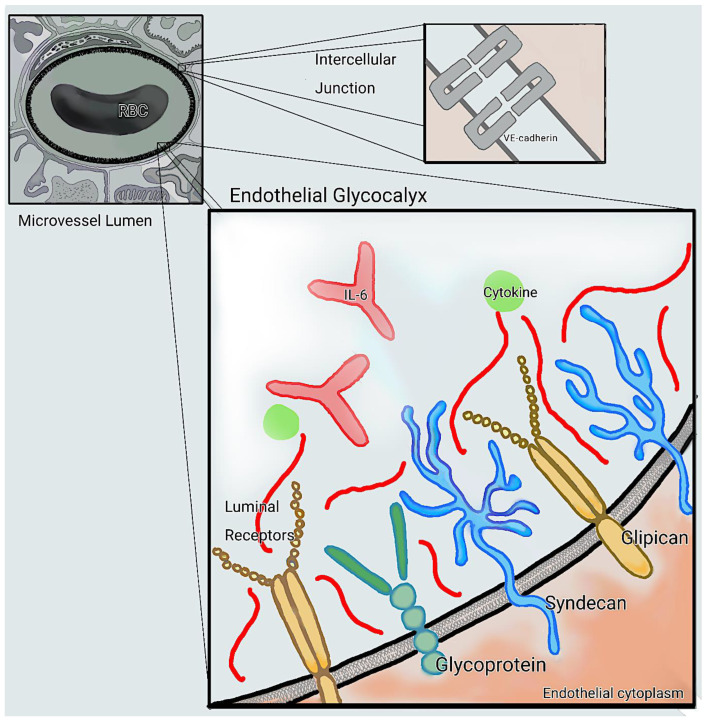
The endothelial glycocalyx of the microcirculation. Red blood cells (RBC) are repelled by the charge created by the proteins of the endothelial glycocalyx. Shear force of RBC in blood maintains integrity of endothelium and its glycocalyx. Syndecans, glipicans and glycoproteins (heparan sulfate and hyaluronic acid) bind water and albumin to create the repellent charge to maintain colloid pressure in the lumen of the vessel. Vascular endothelial cadherin forms bonds at tight junctions that are attacked by inflammatory cytokines, causing escape of fluid into the interstitial space in sepsis.

**Table 1 microorganisms-11-01119-t001:** Immunomodulation and current Immunotherapy clinical trials from ClinicalTrial.gov (Last accessed: 27 January 2023).

Conditions	Therapeutics (s)	Phase	Patients [n]	Trial Identifier	Status
**Severe Sepsis with Septic shock**	Two dosing frequencies of recombinant Interleukin-7 (CYT107) treatment to restore absolute lymphocyte counts in sepsis patients; IRIS-7B	Phase 2	27	NCT02640807	Completed
**Sepsis and Septic Shock**	Effects of Interferon-gamma on Sepsis-induced Immunoparalysis,	Phase 3	4	NCT01649921	Completed
**Severe Sepsis and Septic Shock**	PD-1/PD-L1 pathway inhibition in sepsis, BMS-936559	Phase 1b/2a	35	NCT02576457	Terminated
**Septic Neonates with Neutropenia**	Macrophage colony stimulating factor (GM-CSF) in septic neonates with neutropenia	Phase 1	280	ISRCTN42553489	Completed
**Septic Shock**	Allogeneic mesenchymal stromal cells (CISS)	Phase 1/Phase 2	9	NCT02421484	Completed
**Sepsis and Macrophage Activation Syndrome**	Treatment with recombinant human interferon-gamma or anakinra validation (PROVIDE)	Phase 2	36	NCT03332225	Completed
**Pediatric Sepsis-induced MODS**	GM-CSF for Reversal of immunoparalysis	Phase 4	120	NCT03769844	Active, not Recruiting
**Sepsis**	Drug: Anakinra or rhIFNγ adjunctive immunotherapy (ImmunoSep)	Phase 2	280	NCT04990232	Recruiting
**Sepsis**	Long-term Effects of Thymosin Alpha 1 Treatment	Phase 1/Phase 2 Drug interventions have been done in previous clinical studies	900	NCT04901104	Not yet recruiting

**Table 2 microorganisms-11-01119-t002:** Definitions of phenotypes and the study of omics.

**Phenotype**	The observable traits or characteristics of an organism governing morphology, development, behavior, and properties, resulting from the interaction of its genome with its environment.
**Endotype**	A link to a single molecular mechanism and those that share etiological and pathogenic pathways with nonlinear dynamic interactions that may or may not be present in all patients, or in each patient at all time points.
**Omics**	Characterisation of the biological signal in respect of a disease or subtype of disease process.
**Metabolomics**	Study of the metabolic substrates produced and their timing during a disease or pathological process.
**Proteomics**	Description of the nature, quantity and timing of proteins produced during a disease or pathological process.
**Genomics**	Focus on the structure, function, evolution and editing of the DNA comprising the complete set of genes of an organism.
**Transcriptomics**	Examination of the changes in transcription of RNA information coded in an organism’s DNA that is present in a sample (a cell, tissue, or organ) at a given time.

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
