# Peer review of "Multi-Omics Endotypes in ICU Sepsis-Induced Immunosuppression"

_microorganisms, 2023, doi:10.3390/microorganisms11051119_

Round 1

Reviewer 1 Report

In this review an extensive and detailed analysis is presented on the immunological decoding of sepsis and its correlation with targeted immune-therapeutic approaches. It is described how omics-analysis and the reveal of specific, sepsis-related endotypes could allow a more personalized therapeutical approach of sepsis patients. The manuscript is well written and is almost ready to be published in its current form. I have two comments:

1.       The term “severe sepsis” in the title could be considered a reference to the old sepsis-definition. I would recommend to change. It could be changed for example into “sepsis” to meet the sepsis-3 criteria or into “severe sepsis incident”.

2.       Line 544: Kindly add reference to support the statement.

Author Response

Response to Reviewer 1: Thank you for your considerate assessment of the literature analyzing the potential immunological decoding of sepsis and its correlation with targeted immune-therapeutic approaches. Your insights and comments allowed to follow diagnostic accuracy when it came to the title and to reference all past literature necessary when referencing mortality metrics. 

  1. The term “severe sepsis” in the title could be considered a reference to the old sepsis-definition. I would recommend changing. It could be changed for example into “sepsis” to meet the sepsis-3 criteria or into “severe sepsis incident”.

Answer: We agree the word severe has been deleted and the title shortened. Moreover, we have carefully amended all the “severe” sepsis to sepsis thorough the text. 

  1. Line 544: Kindly add reference to support the statement.

Answer: Thank you for noting the reference to this statement was missing in the text. I have addressed this accordingly and added this reference:

Liu VX, Fielding-Singh V, Greene JD, Baker JM, Iwashyna TJ, Bhattacharya J, Escobar GJ. The Timing of Early Antibiotics and Hospital Mortality in Sepsis. Am J Respir Crit Care Med. 2017 Oct 1;196(7):856-863. doi: 10.1164/rccm.201609-1848OC. PMID: 28345952; PMCID: PMC5649973. – in Line 708

Reviewer 2 Report

The authors provided an interesting review of sepsis endotypes identified by integrative multi-omics approach. However, this manuscript is characterised by several inaccuracies that warrant further clarification. Honestly, I do not really understand what the authors mean with "immune infiltration". Accordingly, they are kindly asked to define this expression. 

The introduction is incredibly long and quite misleading. Moreover, the authors reported a case scenario as example, which appears confusing in this context. I agree with the author that defined the sepsis 3 definition criteria as delayed in the diagnosis and consequent management of sepsis. However, these criteria are pragmatic and easy to use in large scale. Do they have some alternative? I don't think so. Omics is interesting but there are no data that support its use in daily clinical practice. Moreover, tests assessing the "omics" characteristics of patients are time consuming and may end up with delayed sepsis diagnosis again. 

The other sections of the manuscript are quite informative although I suggest to organise them in order to make the reader aware of what remains confined to the research world and what is available in clinical practice and is ready for clinical use. Moreover, it would be nice to read about drugs apart from steroids and vasopressors that may have some potential immunomodulating application in clinical practice. Finally, figure resolution is poor and warrants to be improved.

Author Response

Response to Reviewer 2:

The authors provided an interesting review of sepsis endotypes identified by integrative multi-omics approach. However, this manuscript is characterised by several inaccuracies that warrant further clarification. Honestly, I do not really understand what the authors mean with "immune infiltration". Accordingly, they are kindly asked to define this expression. 

Response: We apologise for this and immune infiltration is deleted.

The introduction is incredibly long and quite misleading. Moreover, the authors reported a case scenario as example, which appears confusing in this context. I agree with the author that defined the sepsis 3 definition criteria as delayed in the diagnosis and consequent management of sepsis. However, these criteria are pragmatic and easy to use in large scale. Do they have some alternative? I don't think so. Omics is interesting but there are no data that support its use in daily clinical practice. Moreover, tests assessing the "omics" characteristics of patients are time consuming and may end up with delayed sepsis diagnosis again. 

Response: We apologize for the long and unfocussed introduction. We have completely fully rewritten the introduction to cover 3 parts: multi-omics data, clinical criteria and the multimodal approach to septic patients with the goal of a narrative review.

The other sections of the manuscript are quite informative although I suggest to organise them in order to make the reader aware of what remains confined to the research world and what is available in clinical practice and is ready for clinical use. Moreover, it would be nice to read about drugs apart from steroids and vasopressors that may have some potential immunomodulating application in clinical practice.

Response: We appreciate this comment and the sections have been reordered as follows:

2a. The identification of deleterious neutrophil states and altered granulopoiesis in sepsis –

2b. Transcriptomic driven endotype in sepsis immunocompromised patients –

2c. Transcriptomic landscape of chronic critical illness in late sepsis –

3a. Sepsis induced metabolic changes and microcirculatory damage –

3b. Biomarkers of endothelial damage to assess treatment response in critical illness –

4a. Hepatic injury dysregulation in sepsis –

4b. Sepsis induced cardiomyopathy gene modular features –

5a. Transcriptomic and immunological metrics guiding immunotherapy –

5b. Pharmacologic Immune Modulation in Sepsis -

In the last section, we have cited some promising therapies with mechanisms that would potentially provide improving in patients’’ outcomes.

Finally, figure resolution is poor and warrants to be improved.

Response: Figure resolution has been amended. Figure 1 has been enlarged and applied higher resolution with legend We hope this new figure resolution satisfies the reviewer’s request.

Reviewer 3 Report

This is a comprehensive review over the Integrative multi-omics of immune infiltration and  immunosuppression in severe sepsis and septic shock.  This review provides the history and updated information in this field. Only minus concerns:

1. Figure 1 needs high resolution and large font for text. Also needs detailed legend.

2. Line 640: w/ shall be changed.

3. Better to give more subtitles, particularly in section 1 and section 3

Author Response

Response to Reviewer 3:

This is a comprehensive review over the Integrative multi-omics of immune infiltration and  immunosuppression in severe sepsis and septic shock.  This review provides the history and updated information in this field. Only minus concerns:

Response: Thanks for this feedback.

  1. Figure 1 needs high resolution and large font for text. Also needs detailed legend.

Response: Figure 1 has been enlarged and applied higher resolution with legend.

Line 640: w/ shall be changed.

Response: This has been amended in the text as per request.

Better to give more subtitles, particularly in section 1 and section 3

Response: As requested by reviewer2 and yourself, the sections have been reordered as it is answered to Reviewer 2 and yourself.

Round 2

Reviewer 2 Report

The authors provided an updated version of this manuscript, which has significantly improved although it deserves further revision.

The introduction appears incomplete. Probably you should move the lines 115-116 after the lines 116-121 otherwise it looks like that something is missing. In the light of this view, you should improve the line 115-116 in order to explain how you would overcome the issues reported in lines 116-121 with the present manuscript.

You should mention the theoretic role of each biomarker (e.g. HLA-DR) in order to make the reader able to understand the reason why it had been investigated in certain study and the role it may play in the development of sepsis phenotypes. In the light of this view, describe what MALS and SICM mean and how they are currently diagnosed.

Line 269: "due to"...what? Probably you should remove brackets. 

Improve the paragraph on steroids as these drugs are widely used in clinical practice. What do they target? How do they act? Please, discuss their role in community acquired pneumonia. Lines 747-750 lack of references. Do they report authors' hypothesis?

Again, say something more on immunoglobulin (type, dose, timing, target)

Shorten the conclusion section

Describe SRS2 profile as you reported only SRS1 characteristics and some comparison with SRS2. 

Report what MARS means.

Author Response

The authors provided an updated version of this manuscript, which has significantly improved although it deserves further revision.

Response: Thanks for the revision done to improve our manuscript. It is appreciated.

The introduction appears incomplete. Probably you should move the lines 115-116 after the lines 116-121 otherwise it looks like that something is missing. In the light of this view, you should improve the line 115-116 in order to explain how you would overcome the issues reported in lines 116-121 with the present manuscript.

Response: We agree and the whole paragraph has been rewritten as follows:

Previous studies aimed to identify sepsis endotypes by analyzing the behavior of monocyte HLA-DR (mHLA-DR) expression during the first week after sepsis onset. The results showed that two-thirds of septic patients exhibited low or decreasing mHLA-DR expression, while in the remaining patients, mHLA-DR expression increased. Bodinier et al. [13]. Discovered that measuring mHLA-DR expression on the first and third day after sepsis onset is sufficient for early risk stratification of sepsis patients. This finding may help clinicians identify patients who are at higher risk of developing complications and who may require more intensive treatment. Overall, this study highlights the importance of monitoring mHLA-DR expression in septic patients and suggests that this biomarker may be a useful tool for identifying sepsis endotypes and for early risk stratification. However, further research is needed to validate these findings and to determine the clinical implications of mHLA-DR monitoring in sepsis management.

You should mention the theoretic role of each biomarker (e.g. HLA-DR) in order to make the reader able to understand the reason why it had been investigated in certain study and the role it may play in the development of sepsis phenotypes. In the light of this view, describe what MALS and SICM mean and how they are currently diagnosed.

Response: Reviewer is right and we have introduced the topic before the two studies describing the HLADR and MACS. We have included the following paragraph:

HLA-DR, or human leukocyte antigen-DR, is a molecule that is expressed on the surface of certain immune cells, including monocytes, macrophages, and dendritic cells. HLA-DR plays a critical role in the immune system's ability to recognize and respond to foreign invaders, such as bacteria and viruses. In sepsis, HLA-DR expression is often decreased, which is thought to reflect a state of immune suppression that can contribute to the development of sepsis and its associated complications. This immune suppression is referred to as immune paralysis, and it can make it more difficult for the body to fight off the infection and can increase the risk of secondary infections and other complications. Similarly, in macrophage activation-like syndrome (MALS), which is a rare but poten-tially life-threatening condition characterized by a dysregulated immune response, HLA-DR expression may also be decreased. This dysregulation can result in the over-production of pro-inflammatory cytokines and other immune molecules, leading to widespread tissue damage and organ dysfunction. Overall, monitoring HLA-DR ex-pression can provide insights into the immune system's ability to respond to infections and other challenges, and may have implications for the diagnosis, treatment, and management of a range of conditions, including sepsis, MALS, and immune paralysis. – Line 112-139

Line 269: "due to"...what? Probably you should remove brackets. 

Response: Agreed and done.

Improve the paragraph on steroids as these drugs are widely used in clinical practice. What do they target? How do they act? Please, discuss their role in community acquired pneumonia. Lines 747-750 lack of references. Do they report authors' hypothesis?

Response: We imagine that the reviewer is refereeing not to lines 747 as this is not within the text. We have however introduced the topic of steroids from a more basic perspective and added previous reviews done in patients with CAP as follows:

Steroids are a class of drugs that have anti-inflammatory and immunosuppressive effects. They are widely used in clinical practice to treat a variety of conditions, including autoimmune diseases, allergic reactions, and certain types of cancer. Steroids work by binding to specific receptors in the body, such as the glucocorticoid receptor (GR), and modulating gene expression and cellular signaling pathways. This can lead to a decrease in the production of pro-inflammatory cytokines and other immune molecules, as well as a decrease in the activity of immune cells, such as T cells and B cells. In the context of community-acquired pneumonia (CAP), steroids have been studied as a potential ad-junctive treatment to antibiotics, with the aim of reducing inflammation and improving outcomes. Several clinical trials have investigated the use of steroids in CAP, and while the results have been mixed, some studies have suggested that steroids may have a beneficial effect in certain patient populations. A recent review paper analyzed sys-tematically all published randomized controlled trials involving patients with severe CAP found that the use of steroids was associated with a significant reduction in mortality and length of hospital stay, as well as a decreased risk of treatment failure. However, it should be noted that the benefits of steroids were primarily observed in patients with severe CAP and those who required mechanical ventilation or vasopressor support. Overall, while steroids may have a role in the management of severe CAP, their use should be considered on a case-by-case basis, considering the potential risks and benefits. It is important to note that steroids can have significant side effects, such as increased risk of infection, hyperglycemia, and gastrointestinal bleeding, among others.  – Line 599-619

Again, say something more on immunoglobulin (type, dose, timing, target)

Response: We have added a paragraph as requested by the reviewer with brief recommendations as follows:

The use of immunoglobulin therapy for sepsis is still a matter of debate, and there is no clear consensus on the type, dosing and timing. The most used types of immuno-globulin for sepsis are intravenous immunoglobulin (IVIG) and IgM-enriched formulations. The optimal dose of IVIG for sepsis is still unclear. Some studies have used a dose of 0.5-1 g/kg, while others have used higher doses (up to 2 g/kg). The dose may also depend on the patient's age, comorbidities, and the severity of sepsis. The optimal timing of IVIG administration for sepsis is also unclear. Some studies have suggested that early ad-ministration of IVIG (within the first 24-48 hours of sepsis) may be more effective, while others have shown benefit with later administration (after 48 hours). – Line 637-645

Shorten the conclusion section

Response: Done

Describe SRS2 profile as you reported only SRS1 characteristics and some comparison with SRS2. 

Response: We agree and the paragraph in page 12 has been moved to page 5 to introduce the SRS1 and 2 concept.

Report what MARS means.

Response: We apologise and a paragraph has been inserted to explain where MARS term came from as follows:

Scicluna et al described four different endotypes upon ICU admission for septic pa-tients using a discovery and validation cohort under the acronym of Molecular Diagnosis and Risk Stratification of Sepsis (MARS) Consortium
